# Spectral Properties Echoing the Tautomerism of Milrinone and Its Application to Fe^3+^ Ion Sensing and Protein Staining

**DOI:** 10.3390/bios12100777

**Published:** 2022-09-20

**Authors:** Hanming Zhu, Pan Ma, You Qian, Jiaoyun Xia, Fuchun Gong, Lusen Chen, Lujie Xu

**Affiliations:** College of Chemistry and Chemical Engineering, Changsha University of Science and Technology, Changsha 410114, China

**Keywords:** milrinone, inter-ESPT fluorophore, tautomerism, Fe^3+^ sensing, protein staining

## Abstract

Knowledge on the spectral properties of the tautomers of milrinone (MLR) in solvents and solid-state, as well as under light conditions is of critical importance from both theoretical and practical points of view. Herein, we investigated the spectral properties of MLR in different conditions using UV-Vis and fluorescence spectroscopies. The experimental results demonstrated that MLR can undergo the tautomerization reaction induced by solvent polarity, light and pH, eliciting four tautomeric structures (enol, keto, anion, and cation forms). The interesting multi-functional groups in MLR enable it to coordinate with metal ions or to recognize gust molecules by H-bonding. In the use of MLR as an excited-state intermolecular proton transfer (inter-ESPT) fluorescent probe, a highly sensitive and selective analysis of Fe^3+^ was developed, which offered a sensitive detection of Fe^3+^ with the detection limit of 3.5 nM. More importantly, MLR exhibited the ability of anchoring proteins and led to the recognition-driven turn-on inter-ESPT process, highlighting the potential for the probe to image proteins in electrophoresis gels. The spectral experimental results revealed the possible degradation mechanism, so that we can better understand the side effects of oral preparations. The use of the available drug as an inter-ESPT fluorescent probe is simple and accurate, providing a good method for Fe^3+^ ion sensing and protein staining.

## 1. Introduction

Milrinone (MIR), as a selective inhibitor of the phosphodiesterase-3-enzyme (PDEIII), is an effective cardiotonic drug and is used for the treatment of heart failure [1,2]. MIR exhibits an inotropic and vasodilatory effect by suppressing the cAMP degradation. Similar to the other phosphodiesterase (PDE) inhibitors, MLR are often followed periodically due to their side effects, such as hemolysis, elevated liver enzymes, thrombocytopenia, and electrolyte disturbances [3,4]. Many studies have confirmed that the mean decay lifetime of MIR is three times higher than that in the patients with acute kidney injury [5]. Patients with long time exposure to MIR can experience tachyarrhythmias and sudden cardiac death. High doses of MIR may cause hypotension and syncope. In addition, excessive MIR increases the possibility of side effects observed in the patient and interaction with other drugs [6,7,8]. Therefore, the accurate analysis of the tautomers of MIR is of great significance for the quality control in drug production and clinical use.

The pH of the medium can seriously affect the structural form of MLR [9] (Figure 1). At a neutral pH, under physiological conditions, the keto structure predominates. Following exposure to an acidic environment, it appears as a cation. Both the keto and cationic structures are the active forms of the MIR inotropic agent [10,11]. The efficient non-radiative relaxation for the tautomeric structures of MIR is interpreted in the fields of H-bonding, charge-transfer (ICT), and the twisting motion of the functional group [12]. It is these mechanisms that make it have the critical features for the biologic activity of keto and cation structures [13]. However, a great deal of investigations into the oral use of MLR revealed that thrombocytopenia is related to the adverse side effects of MLR. It is clear that the active forms of MLR employed for the therapy of cardiac disease are keto (K) and cation patterns [14,15]. Although the biologic activity of MIR depends on its ground state, light may have a great influence on its tautomerization and it is worthwhile to study its photodynamics. Actually, some adverse side effects are associated with the tautomers of MLR resulting from oral preparation processing or gastrointestinal digestion [16,17]. The data of the short time scale results may allow for a better understanding of its interaction with solvents, and for elucidating possible sources of the main degradation channels of MLR. Consequently, the monitoring of tautomers of MLR is also of great urgency.

Besides being a drug, MLR is an interesting multi-functional compound which is representative of typical excited-state intermolecular proton transfer (inter-ESPT) fluorogens [18,19]. The inter-ESPT fluorogens have been used as fluorescence probes to monitor the solvation dynamics and the polarity of water in biological systems (e.g., proteins) [20], as ultraviolet stabilizers [21], and as transition metal ion sensors [22]. MLR has a 2-pyridone core and exhibits two stable tautomeric structures: the enol form (HPy) and the keto form (Py). The keto-enol forms of tautomerization of 2-hydroxypyridine derivatives have been broadly studied experimentally and also theoretically as the simplest systems of intramolecular and intermolecular proton-transfer reactions [23,24,25]. MLR has another pyridine-N in addition to the inter-ESPT system, endowed with the ability of hydrogen bonding and metal ion coordination.

Ferric ion (Fe^3+^) acts as a crucial and plentiful metal ion in biosystems and in the environment [26]. It conducts vital functions in several biological processes, such as enzyme catalysis, cell metabolism, as well as oxygen and electron transport [27]. Excessive or lack of Fe^3+^ will lead to numerous illnesses including anemia, Parkinson’s syndrome, and Alzheimer’s disease [28,29]. To meet the challenge caused by abnormal Fe^3+^ concentrations, it is very important to monitor iron concentrations in ecological water. Consequently, an important consideration has been dedicated to develop diverse methodologies to track Fe^3+^ quantitatively. Fluorescent probe-based analysis has been extensively employed because of the advantages of high specificity, simplicity, prompt response, and so on [30,31].

In this work, the optical characters of MLR were thoroughly investigated using UV-Vis and fluorescent spectroscopies. By using MLR as an inter-ESPT fluorescent probe, a sensitive and high selective analysis of Fe^3+^ was established. Under the induction of Fe^3+^ ions, the inter-ESPT reaction of the fluorescent probe was suppressed due to the coordination of Fe^3+^ with inter-ESPT system, which led to the change in the fluorescence intensity. Meanwhile, the color of the resulting solutions gradually turned to pink. Interestingly, MLR can not only anchor onto its target protein PDEIIIA, but also attach on the proteins in electrophoresis gel, revealing that MLR can sense proteins and be used as a protein stain. The theoretical calculation results support the spectral experiments, revealing the inter-ESPT reaction and the affinity of MLR for proteins. Therefore, the ESIPT determination and naked-eye observation of Fe^3+^ ions can be realized. The real samples were determined using the proposed MLR-based method, indicating satisfactory recoveries with reasonable RSDs. The MLR-based inter-ESPT fluorescent probe is simple and accurate, proposing a good method for the monitoring of Fe^3+^ions and protein sensing.

## 2. Experimental Section

### 2.1. Reagents and Instruments

Milrinone (99.5%, MLR) was obtained from Hunan Hengsheng Pharmaceutical Co., Ltd. (Changsha, China). All the other chemicals and solvents were used as received from commercial sources. The stock solution of MLR (10 mM) was prepared by dissolving an appropriate amount of MLR (2.11 g) in 10 mL DMF and then the volume of 100 mL was set with a DMF−HEPES buffer solution (3/7, *v*/*v*, 10 mM, pH 7.0). The working solutions were prepared by diluting the corresponding stock solutions to an appropriate volume with the same DMF−HEPES mixed system whenever required. The expressed polypeptides MarTX and Ecallantide were obtained from the State Key Laboratory of Bioorganic and Natural Products Chemistry, Shanghai Institute of Organic Chemistry, Chinese Academy of Sciences (Shanghai, China).

The fluorescence spectra were recorded on an F-7000 fluorescence spectrophotometer (F-7000, Hitachi, Tokyo, Japan). UV-vis absorption measurements were performed using a Cary 60 UV-vis spectrophotometer (Agilent Technologies, Mulgrave, VIC, Australia). 1HNMR spectra were acquired with a Bruker A VB-400 MHz NMR spectrometer (Bruker BioSpin, Fällanden, Switzerland). Fourier transform infrared (FT-IR) spectra in KBr were obtained with a WQF-510 FT-IR spectrometer (Beijing Rayleigh Analytical Instrument Co., Ltd., Beijing, China).

### 2.2. Measurement Procedure

A 20 μL MLR stock solution was diluted into 5 mL with a HEPES buffer solution (10 mM, pH 7.2) in a 10 mL-graduated tube. A total of 10 mM Fe^3+^ standard solution was prepared by dissolving an appropriate amount of Fe_2_(SO_4_)_3_ (4.00 g) in 25 mL deionized water containing 0.1 mL H_2_SO_4_, then diluted with water to 1000 mL. The tested Fe^3+^ standard solution (or sample solutions) was then added into the test tube. After setting to 10 mL with the same HEPES buffer solution, the mixture system was shaken thoroughly to react for 20 min at room temperature prior to the fluorescence measurements. Simultaneously, the controls without Fe^3+^ standard solutions or sample solutions were acquired from corresponding solvents. Final, the fluorescence emission at 400 nm for the test solution and the blank reagent were directly recorded on an F-7000 fluorescence spectrophotometer with an excitation wavelength of 323 nm.

### 2.3. Preparation of Real Samples

The tap water was collected from our laboratory. The well water and river water were collected from the waterway in our campus and Xiangjiang River (Changsha, China), respectively. The water samples (100 mL, three parallel samples) were filtered through a 0.45 μm membrane filter, and their pH value was adjusted to 7.0 by HEPES (10 mM). The resulting samples were then used for determination.

### 2.4. Protein Staining

All protein samples were obtained from the State Key Laboratory of Bioorganic and Natural Products Chemistry, Shanghai Institute of Organic Chemistry, Chinese Academy of Sciences (Shanghai, China). Binding of MLR to the expressed proteins (Ecallantide, MarTX and their inclusions) was carried out on electrophoresis gels. According to the conventional protein separation method, the expressed proteins and their inclusions were first separated by SDS−PAGE. After staining with MLR, the electrophoresis gels were decolorized in water for 1 h, followed by in-gel fluorescence scanning (254 nm Argon excitation, 520 BP 40 emission filter, Typhoon 9410 imaging system, GE Healthcare). Coomassie brilliant blue R 250 was used as referenced stainer to stain proteins in the electrophoresis gels. The MLR-based staining solution was prepared as follows: An appropriate amount of MLR (0.021 g) was first dissolved in 15 mL DMF and the MLR solution was then diluted to 500 mL with a DMF−water mixed system (3/7, *v*/*v*), obtaining a 20 mM MLR staining solution. The 0.25% Coomassie brilliant blue R 250 was prepared by dissolving 0.25 g Coomassie in 45 mL methanol, followed by adding 45 mL water and 10 mL glacial acetic acid to the Coomassie-methanol solution.

### 2.5. Computational Methods

The geometric configurations in the ground states of the enol and keto tautomers of MLR were optimized using the density functional theory (DFT) and time-dependent density functional theory (TDDFT) methods, respectively. Using the B3LYP method and the 6-311++G(d,p) higher order basis set all the calculations were made, which were proven to be very reliable for the tautomers. The geometrical parameters such as HOMO and LUMO were calculated by the optimized structure. All calculation results in this work were obtained from the Gaussian 03 W program package on a personal computer. The enol and keto tautomer molecules were docked with BSA protein using Chimera 1.14 and Autodock vina software.

## 3. Results and Discussion

### 3.1. Spectroscopic Characteristics of MLR

The UV-Vis spectrum of MLR was measured in DMF−HEPES mixed system (3/7, 10 mM, pH 7.0). As shown in Figure 2a, MLR exhibited two strong absorption bands at 280 nm and 332 nm, respectively. Thus, we attributed the highest energy bands to the n−π* transition and the lower one to the π−π* transition, respectively. The fluorescence spectra of MLR in DMF−HEPES indicate that three emission peaks can be observed (Figure 2b). The strongest peak at 403 nm was appointed to the normal fluorescence and the second one located at 476 nm corresponded to tautomeric fluorescence, the last peak was assigned as stacked dimeric emission at 513 nm. 

Generally, the UV-Vis absorption and fluorescence spectra of inter-ESPT molecules are seriously affected by the solvents, and the sensitivity and selectivity of the probes to the analyte varies depending on the solvents. In order to evaluate the effect of solvents on the optical properties of MLR, we recorded the absorbance changes of MLR in different solvents. As shown in Figure 2c, the spectroscopic patterns of MLR in different solvents were similar to those in the DMF−HEPES mixed system, except for acetic acid. However, their absorbencies changed. In addition, a new absorption band at 426 nm can be observed in acetic acid, which may have been caused by pH change. The fluorescence measurements of MLR in different solvents were also carried out. As shown in Figure 2d, one obvious mission band of MLR appeared and was accompanied by an obvious bathochromic shift (with the maximum peaks at about 400 nm) sustaining their absorption spectra, which indicates that it was difficult for the inter-ESPT process to occur in the tested solvents, except for acetic acid, so that the strong emission was ascribed to keto-tautomeric form. This different behavior of MLR may have arisen from the excited-state double proton transfer (ESDPT) process mediated by acetic acid and the give off of tautomeric fluorescence, which has been demonstrated in the literature [32]. Moreover, the dual emission of MLR manifested a solvent-dependent PL property, which is similar to what was reported for 2-pyridinone derivatives based on ESIPT [33].

Fluorescence quantum yields (Φ_f_) values were measured using quinine sulfate (0.05 M) in 0.5 M H_2_SO_4_ as a quantum yield standard sample (Φ_f_ = 0.55). The fluorescence spectra were recorded at 25 °C with a solution absorbance < 0.05. The results were determined in six different solvents. MLR had the Φ_f_ value of around 0.55, displaying obvious solvent-dependence. The highest value was up to 0.58 in tetrahydrofuran, while the lowest one was 0.36 in dichloromethane. Since the inter-ESIPT mechanism is quite dependent on the solvent properties, this variation on the Φ_f_ was expected. This Φ_f_ value of MLR was moderate.

### 3.2. Spectroscopic Response of MLR to Fe^3+^

Considering the multiple coordination sites, we investigated the sensing properties of MLR to Fe^3+^. Figure 3a presents the absorption spectra of MLR and the MLR with the addition of Fe^3+^ ions. It shows that the absorption peak of MLR at 332 nm decreased appreciably with the addition of Fe^3+^ ions, while the absorption band at 280 nm was significantly enhanced. These results underwent that the conversion of the free MLR to the corresponding MLR–Fe^3+^ complexes carried out. The absorption change at 332 nm could be attributed to the suppression of the ESIPT reaction in the MLR–Fe^3+^ complexes. The enhanced absorption may be attributed to the absorption of the MLR–Fe^3+^complexes at 280 nm. 

The fluorescence spectra of MLR in the same DMF−HEPES mixed system with the addition of Fe^3+^ indicated that the peak at 400 nm was lowered appreciably (Figure 3b). Consequently, by using MLR as a probe for Fe^3+^, the fluorescence titration experiments were carried out. The results are presented in Figure 3c,d. It is clear that the normal emission intensity of MLR at 400 nm decreased gradually with the increase in the Fe^3+^ concentration. With the addition of 50 μM Fe^3+^, a new fluorescence band at 440 nm emerged, which was attributed to the binding–saturation of the coordination atoms in MLR. The detection limits of MLR were calculated to be 3.5 nM for Fe^3+^. 

MLR bears a D-A system and exhibited typical inter-ESPT features in protic solvents: a short-wavelength emission at 400 nm and a long wavelength fluorescence band at 476 nm, corresponding to the keto and enol tautomers, respectively. The two tautomers of MLR were then testified using HRESI−MS. As shown in Figure 4, the sample showed two components at 2.730 and 3.594 min after high performance liquid separation using an acetonitrile−PBS buffer (pH 7.2) as the eluent. The mass spectrogram revealed that two components corresponded to the enol and keto form of MLR, respectively.

### 3.3. Effect of pH on the Tautomerization of MLR

We also investigated the effect of pH on the fluorescence behavior of MLR. As shown in Appendix A, the normal fluorescence of MLR increased gradually with the increase in pH value, and the strongest value was realized when the pH of the mixed system (DMF/H_2_O) was up to 7.0. However, the fluorescence spectra of MLR indicated that a new emission band appeared as the pH value lower than 4.5, which was ascribed to the inter-ESPT reaction, giving off tautomeric emission. In order to facilitate the determination of Fe^3+^, the pH 7.0 was taken to examine the response of the probe based on the comprehensive consideration. 

### 3.4. Selectivity of MLR towards Fe^3+^

The response performance of MLR was investigated using fluorescence spectroscopy. In the DMF/HEPES solution (*v*/*v* = 3/7, 10 mM, pH 7.2), MLR gave off the strong normal fluorescence at about 400 nm (*λ_ex_* = 320 nm). However, the C-NH and C=O tautomerization was the predominant decay process for MLR in the excited-state, and this event greatly masked the enol fluorescence at 476 nm. Hence, only a strong blue fluorescence of MLR at 400 nm appeared. Interestingly, in the presence of 20 equiv. of various cations, only the Fe^3+^ ions resulted in an acute fluorescence decrease at 400 nm and were accompanied by an obvious color change, enabling us to distinguish Fe^3+^ by the naked eye (Figure 5). 

The selectivity of MLR towards cations was also evaluated using fluorescence spectroscopy (Figure 5). No significant fluorescence change was observed in the presence of common cations (K^+^, Na^+^, Ca^2+^, Zn^2+^, Mg^2+^, Ba^2+^, Pb^2+^, Ni^2+^, Mn^2+^, Cd^2+^, Cu^2+^, Co^2+^, Al^3+^, and Hg^2+^). The addition of Fe^3+^ in the test system caused obvious change in the normal emissions. When the coexisting metal ions with the same concentration were added to this system, no change in the fluorescence intensity was observed. These results clearly demonstrate that the MLR had a high selectivity to Fe^3+^ compared with most other competitive metal ions.

### 3.5. Detection of Fe^3+^ in Real Samples Using MLR

Considering the sensitive and selective response of MLR to Fe^3+^ ions, the inter-ESPT fluorescence assay was established for the detection of Fe^3+^ using MLR as probes. The normal emission of MLR decreased gradually with the increase in Fe^3+^ concentrations, which was a linear correlation. The detection limit resulted from the formation of MLR−Fe^3+^ complexes was calculated as the 3.5 nM (3σ/s), which was lower than that of the reported fluorescent probe-based method [34,35], displaying a high sensitivity of MLR towards Fe^3+^ ions.

To examine the applicability of this inter-ESPT fluorescent probe in real samples, we determined the Fe^3+^ level in water and soil samples using MLR. The water samples were obtained from the described procedure in the Experimental Section. The real samples were spiked with the prepared standard Fe^3+^ solutions and then determined using the MLR-based probe method. The determination results are listed in Table 1. From this table, we can notice that the recovery studies of the spiked Fe^3+^ containing samples showed satisfactory results (98.3–103.4%) with the reasonable RSD ranging from 3.1 to 3.7%. Our present method seems feasible for the determination of Fe^3+^ in the real samples.

### 3.6. Theoretical Calculation and Analysis

To better understand the spectral features of MLR, the theoretical calculations and discussion were performed. Frontier molecular orbitals (MOs) are often employed for cognizing the charge transfer and distribution of molecular configurations. The MOs of MLR in DMF are depicted in Figure 6a. We evaluated the effect of the highest occupied molecular orbital (HOMO) and the lowest unoccupied molecular orbital (LUMO) on the S1 state of the MLR and its tautomers which are in relation to the two orbitals. Because of that the charge distribution of the HOMO−LUMO of their tautomers are mirror symmetrical, so the electron transition from the S0 state to the S1 state is assigned to the π-π* type. After electron transition from HOMO to LUMO, the tautomers concert a distinctive intramolecular charge transfer (ICT) process, i.e., the charge distribution is also transferred from the hydroxyl nitrogen to the pyridine nitrogen. Therefore, this ICT process leads to the hydroxyl-O becoming more acidic and the pyridine-N becoming more basic. The gap between HOMO and LUMO energy decides the molecular chemical stability. The energy gap between the HOMO and the LUMO of the enol form was 5.00 eV, which was higher that of the enol tauotmers (4.21 eV) and indicated a higher electron conductivity of keto tautomers. This implies that the amino form is mainly in the ground state and the keto form is more prone to electron transfer. Moreover, the molecular dockings were also carried out using the docking software with the method of Auto-dock vina. The PDEIIIA from the PDB database (ID: 7L27) was employed for host molecules. As shown in Figure 6b, both the enol and keto conformers of MLR can bind to the PDEIIIA and tend to ingest into different pockets. The enol tautomers allow the amino acid residues of ASN-681 to be anchored in the specific cavity of PDEIIIA by hydrogen binding, while the keto conformers prefer to attach on the CYS-1073. More importantly, the amino and keto tautomers of MLR can attach to the same PDEIIIA molecule. The results of the molecular docking simulation are consistent with the protein staining experiment, revealing the high affinity of MLR to proteins.

### 3.7. Protein Staining Using MLR 

MLR possesses multiple hydrogen-bond donating and accepting sites and affords the superamolecular recognition. Thus, the expressed proteins MarTX and Ecallantide were selected as the protein models to verify the recognition potential of MLR to proteins. In this experiment, Coomassie brilliant blue R 250 was employed for the referenced dyes to stain MarTX and Ecallantide. First, the MarTX, Ecallantide, and their inclusions were subjected to sodium dodecyl sulfate-polyacrylamide gel electrophoresis (SDS−PAGE) to separate the specific proteins. After staining with MLR, the fluorescence scanning was carried out on the gel bands. As shown in Figure 7a–c, the recognition of MLR to MarTX, Ecallantide and their inclusions really happened, thus resulting in the distinct protein bands. It indicates that all kinds of the proteome can be clearly distinguished by fluorescence imaging with gel imager or ultraviolet scanner. These results confirmed that MLR can recognize proteins and ingest in them. As a traditional protein indicator, Coomassie brilliant blue R 250 is often used to stain the proteins in electrophoretic gels. However, it involves a tedious decoloration procedure. In addition, the usage of MLR is lower (10^−6^ M/L) compared with Coomassie brilliant blue R 250 and shows a clearer staining after decolorization in water for 1 h (Figure 7d–f). 

## 4. Conclusions

In summary, we measured the spectroscopic features of MLR that corresponded to their tautomers in a DMF−water mixed system at varying pH values and in different solvents. Using MLR as an inter-ESPT fluorescent probe, a highly sensitive and selective analysis of Fe^3+^ was established based on the suppression of the inter-ESPT reaction. The determination results of real samples demonstrated that our proposed method had a high selectivity, sensitivitym and simplicity. We also revealed that MLR can anchor the expressed proteins in electrophoresis gels and can be used for protein staining, revealing the obvious advantages of sensitivity and fast decoloration.

## Figures and Tables

**Figure 1 biosensors-12-00777-f001:**
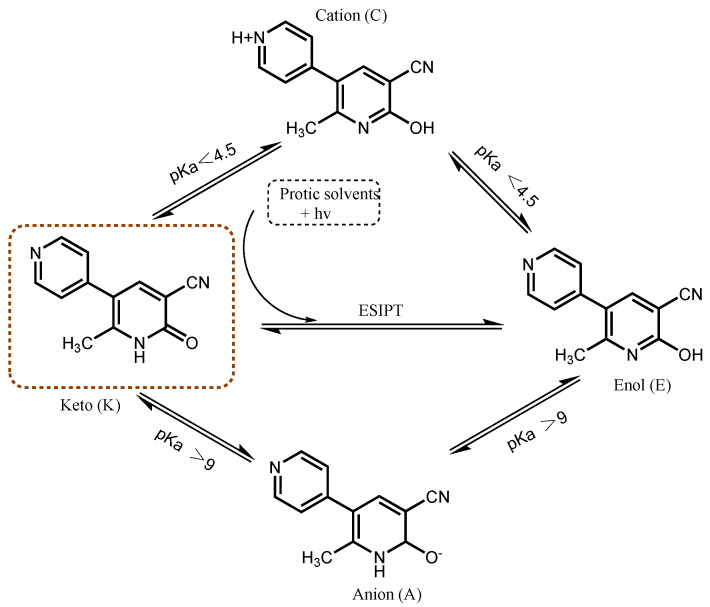
Schematic representation of the possible tautomeric equilibria between the different species of MLR.

**Figure 2 biosensors-12-00777-f002:**
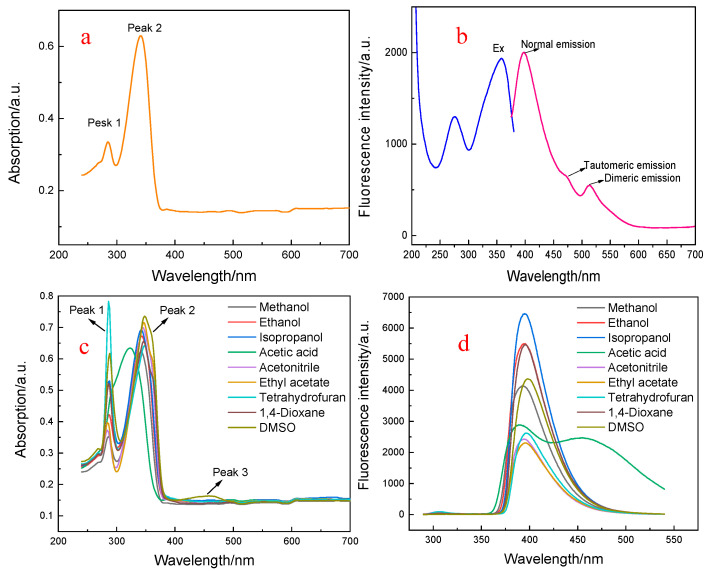
Absorption and fluorescence spectra of MLR. (**a**) Absorption spectrum of MLR in DMF−water mixed system (3/7, *v*/*v*); (**b**) fluorescence spectra of MLR in the same DMF−water system; (**c**) UV-Vis spectra of MLR in different solvents (10 μM); (**d**) fluorescence spectra of MLR in different solvents.

**Figure 3 biosensors-12-00777-f003:**
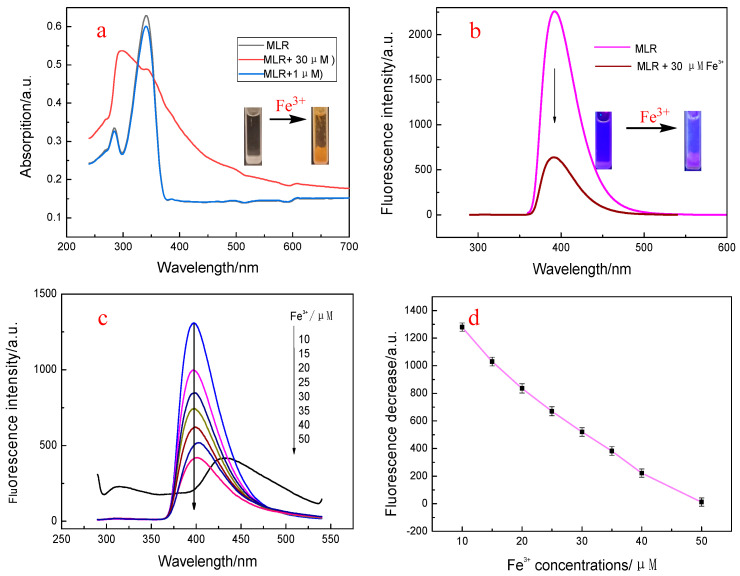
Absorption and fluorescence spectra of MLR with the addition of Fe^3+^. (**a**): Absorption spectrum of MLR in the DMF−HEPES mixed system (3/7, *v*/*v*) upon the addition of Fe^3+^; (**b**): Fluorescence spectra of MLR in the same DMF−water system upon the addition of Fe^3+^; (**c**): Fluorescence spectra of MLR with the addition of different Fe^3+^concentrations; (**d**): linear relationship between ΔF and Fe^3+^ concentration (10~50 μM).

**Figure 4 biosensors-12-00777-f004:**
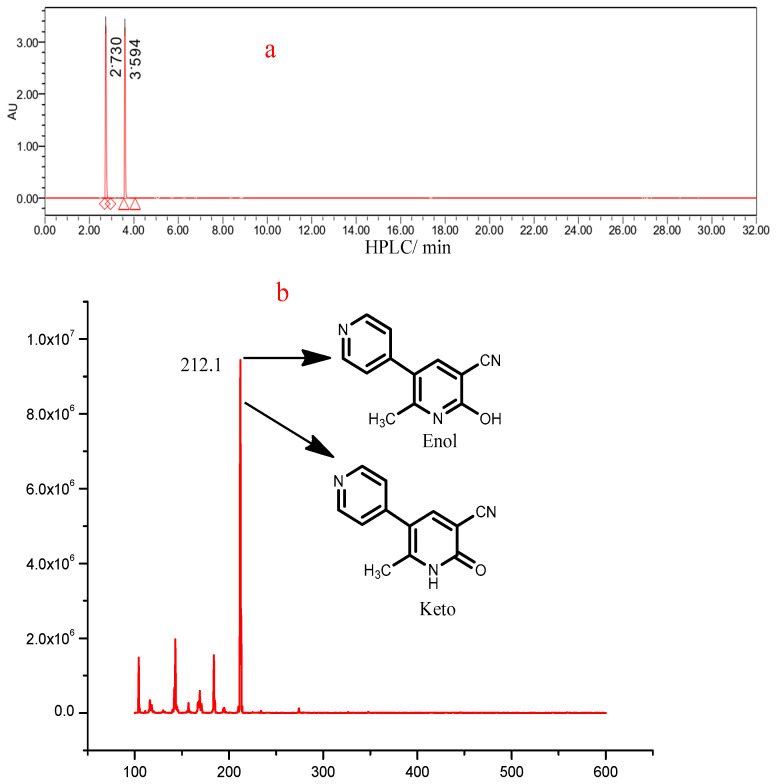
(**a**): HPLC-UV of MLR; (**b**): HRMS of the enol and keto form of MLR.

**Figure 5 biosensors-12-00777-f005:**
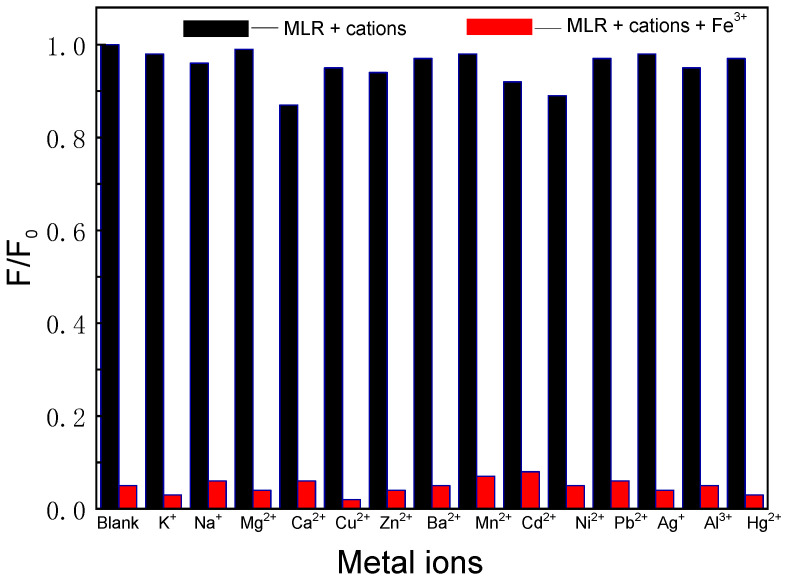
Fluorescence decreases at 400 nm of MLR (2.0 μM) in the presence of different metal ions (20 equiv.) (black); the fluorescence spectra changes of MLR (2.0 μM) with the addition of the metal ions mixture and 40 μM Fe^3+^ ions (red).

**Figure 6 biosensors-12-00777-f006:**
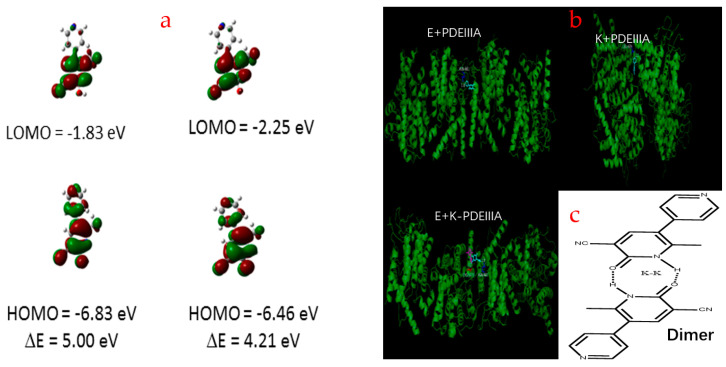
Frontier molecular orbitals (HOMO and LUMO) of the two tautomeric forms of MLR with the energy level (**a**), molecular docking of enol/ketotautomers to PDEIIIA (**b**) and the pattern of the dimers of MLR (**c**).

**Figure 7 biosensors-12-00777-f007:**
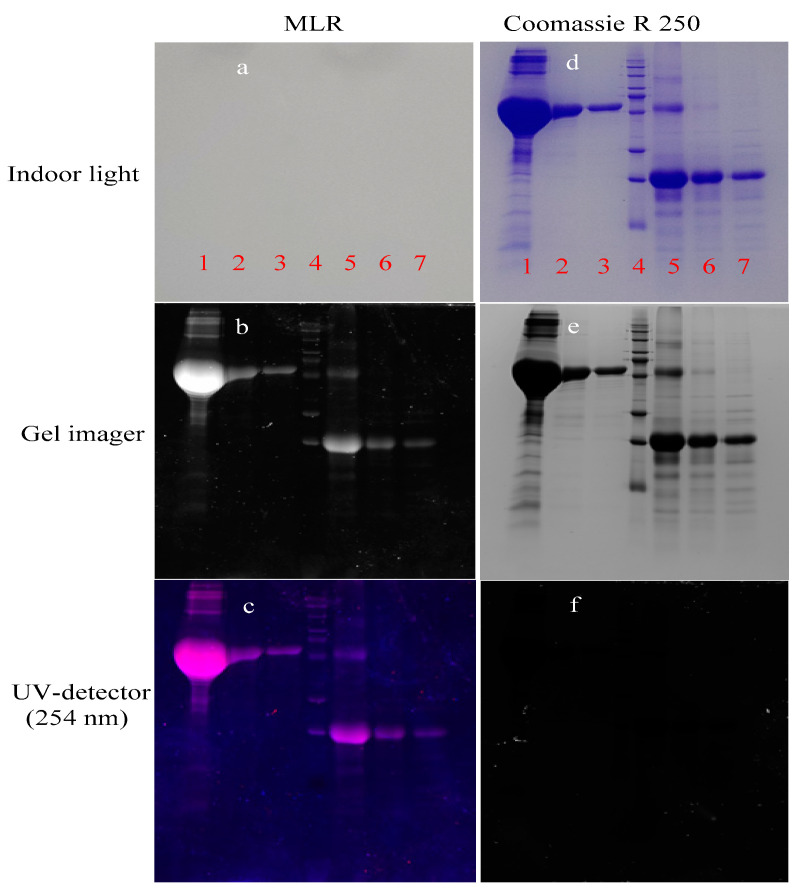
Fluorescence imaging of proteins in electrophoresis gels. (**a**–**c**): The expressed proteins and their inclusions in electrophoresis gels stained with MLR. (Lane 1–7: 1. Bacterial total proteins containing Ecallantide; 2. Supernatant solution containing Ecallantide; 3. MarTX separated by Ni-column and eluted without imidazole; 4. MarTX separated by Ni-column and eluted with 100 mM/L imidazole; 5. MarTX separated by Ni-column and eluted with 500 mM/L imidazole; 6. A total of 100 mM/L Ecallantide; 7. Protein ruler). (**d**–**f**): The same protein samples in electrophoresis gels stained with Coomassie brilliant blue R 250.

**Table 1 biosensors-12-00777-t001:** Analytical results of Fe^3+^ in real samples using MLR as a probe.

Samples ^a^	Added (μM)	Found (μM)	Recovery (%)	RSD (%)
River water	0	1.93	100.1	3.2
	5	8.98	99.7	3.4
	10	17.82	98.5	3.6
Tap water	0	5.21	99.6	3.7
	5	10.42	100.2	3.6
	10	15.45	98.3	3.5
Well water	0	4.93	103.4	3.5
	3	8.81	102.3	3.2
	6	11.02	99.7	3.1

^a^ Average of three determinations.

## Data Availability

Not applicable.

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
