# Peer review of "Spectral Properties Echoing the Tautomerism of Milrinone and Its Application to Fe3+ Ion Sensing and Protein Staining"

_biosensors, 2022, doi:10.3390/bios12100777_

Round 1

Reviewer 1 Report

The manuscript presents the spectral properties of tautomers of milrinone and its application to Fe3+ ion sensing and protein stain. The study is interesting, some questions and errors are listed below:

Abstract:

1.        I suggest deleting “from 0« in the sentence” a highly sensitive and selective analysis of Fe3+ was developed, which offered a sensitive detection of Fe3+in the concentration range from 0 to 50 μM with the detection limit of 3.5 nM.« because you detect the concentration of Fe3+, so they must be present.

2.       Insert a space between Fe3+ and in.

Introduction

1.       Insert a space between Fe3+and concentration.

2.       In the Introduction you need to add the applicability of your research.

Experimental section

1.       Insert a space between number and unit in “20μL MLR stock solution”

2.       Write 3+ in superscript “The tested Fe3+ standard solution”

3.       Please indicate the concentration of the Fe3+ standard solution and write how the dilutions were prepared.

4.       Insert a space between to and 10 in “to10 mL with the”.

Results and discussion

1.       What was the concentration of quinine sulfate?

2.       Figure 3; Why are the concentration of Fe3+ in the absorbance and fluorescence spectra in Figure 3a and 3b not identical? Add a legend in Figure 3c. It is not clear that the black line is 50 μM Fe3+. Explain Figure 3d, at which wavelength the linear relationship was established. Correct the unit of the x-axis. The description of Figure 3d is not correct, the concentration of Fe3+ is in the range from (10-50 μM).

3.       How were the detection limits calculated?

4.       Unify figures 4a, 4b and 4c. In 4c and 4b the x and y-axes are not visible.

5.       Which figure is the Figure 4S? “We also investigated the effect of pH on the fluorescence behavior of MLR. As shown in Figure S4.

6.       Write 3+ superscript in “Upon the addition of Fe3+ in the test system”

7.       In the chapter 3.5 unify the letter size in some sentence the letters are larger.

8.       I suggest to delete “from 0” in the sentence … linearly correlation ranging from 0 to 50 μM

Conclusions

1.       In the conclusion, highlight the results of your study.

Reviewer 2 Report

The sensing property of MIR as a Fe3+ sensor was reported in this manuscript. But I don’t think it is proper to publish this paper based on the following reasons:

1.       In the introduction part, the authors emphasize the side-effects of MIR to the healthy mainly, so it is not proper to use it as Fe3+ sensor for application.

2.       As a sensor with inter-ESPT mechanism, the protonic solvents and aprotic solvents, even the content percentage of the protonic solvents, will affect the sensing properties greatly, the authors did not research and summarize the effect of different kinds of solvents.

3.       In page 5, “Figure 2. ¼¼(d) Fluorescence spectrum and fluorescence spectrum of MLR in different solvents. Inset in d) is the image of the crystals of MLR”. The inset and the list of solvents in Figure 2d are not shown.

4.       In page 5, “As shown in Figure 2d, one obvious mission bands of MLR appeared and accompanied by an obvious bathochromic shift (with the maximum peaks at 400 nm about) against their absorption spectra, which indicate that the inter-ESPT process is difficult to occur in the tested solvents, except for acetic acid, so that the strong emission is ascribed to keto-tautomeric form.” In fact, it is general knowledge that there is a bathochromic shift of the fluorescence spectrum against the absorption spectrum. In addition, the absorption spectrum in 1,4-Dioxane is much different to the absorption spectra in other solvents. The authors should summarize the regularity of the effect of different solvents.

5.       Figure 5 shown the antijam property of MIR as a Fe3+ sensor, but the selectivity of MLR towards cations are also very important, the Figure should be given in the paper.

Reviewer 3 Report

The authors present in this work the spectroscopic data of milrinone (MLR) corresponded to their tautomers in a DMF-water system varying pH and solvents, and also showing the employment of MLR as an inter-ESPT fluorescent probe, highly sensitive to Fe3+. I would like to highlight the great introduction with appropriate bibliography that allow to know more about the field and the literature that support the statements in the manuscript. In the article, the authors carried out a lot of studies with MLR, object of study in this work, with also theoretical calculations to support the experimental data. For all these reasons, I consider that this article is suitable for publication only under minor corrections or suggestions.

1.       In some statements of the article, the “3+” of the cation Fe3+ does not appear as a superindex as in other parts of the article. Please, check it and write all in superindex.

2.       In Figure 2c. What is the explanation of the behaviour in 1,4-dioxane (colour green) with respect the other solvents? In the text the authors comment the different behaviour in acetic acid but I think it could be interested to describe the behaviour in 1,4-dioxane too.

3.       In Figure 2d, why is the different behaviour for pink line? A summary in the graph with the different colour sand solvents should be included to make clearer it for the reader.

4.       Figures 4b and c should be improved because the numbers in the peaks and in y and x axis are difficult to see.

5.       When the authors describe the HOMO and LUMO values, these values should be rounded to the two decimal places. For example, 4.99674 eV should be written as 5.00 eV or 4.21368 eV as 4.21 eV. Please, check it in Figure 6 and in the text.

Round 2

Reviewer 2 Report

This manuscript in the present version is publishable.